# Photobiomodulation Therapy and Pulp-Regenerative Endodontics: A Narrative Review

**DOI:** 10.3390/bioengineering10030371

**Published:** 2023-03-17

**Authors:** Jiawen Yong, Sabine Gröger, Zuping Wu, Sabine Ruf, Yuer Ye, Xiaoyan Chen

**Affiliations:** 1Stomatology Hospital, School of Stomatology, Zhejiang University School of Medicine, Zhejiang Provincial Clinical Research Center for Oral Diseases, Key Laboratory of Oral Biomedical Research of Zhejiang Province, Cancer Center of Zhejiang University, Hangzhou 310003, China; 2Department of Orthodontics, Faculty of Medicine, Justus Liebig University Giessen, 35392 Giessen, Germany

**Keywords:** photobiomodulation therapy, low-level laser therapy, regenerative endodontic procedures, pulp regenerative endodontics, human dental pulp stem cells

## Abstract

Regenerative endodontic procedures (REPs) were used to recover the dental pulp’s vitality in order to avoid the undesirable outcomes of conventional endodontic treatment and to promote dentinal formation, especially for immature permanent teeth. Photobiomodulation therapy (PBMT) exhibits photobiological and photochemical effects for improving the root canal’s environmental conditions by compensating for oxidative stress and increasing the blood supply to implanted stem cells and improving their survival. Basic research has revealed that PBMT can modulate human dental pulp stem cells’ (hDPSCs) differentiation, proliferation, and activity, and subsequent tissue activation. However, many unclear points still remain regarding the mechanisms of action induced by PBMT in REPs. Therefore, in this review, we present the applications of laser and PBMT irradiation to the procedures of REPs and in endodontics. In addition, the effects of PBMT on the regenerative processes of hDPSCs are reviewed from biochemical and cytological perspectives on the basis of the available literature. Furthermore, we consider the feasibility of treatment in which PBMT irradiation is applied to stem cells, including dental pulp stem cells, and we discuss research that has reported on its effect.

## 1. Introduction

The typical method applied in endodontics is root canal treatment (RCT) to treat irreversibly inflamed or necrotic pulp tissue that has been damaged by infectious diseases or trauma [1]. However, this approach has several impairments, such as the possibility of reinfection due to microleakage [2], hypoesthesia, and increased susceptibility to root fracture due to brittleness [3]. To overcome these drawbacks, the purpose of regenerative pulp treatment is to maintain the vitality of dental pulp [4]. Regenerative pulp treatment is an alternative treatment modality during continuous tooth root development and root apical closure in the case of immature permanent teeth [5]. Traditionally, apexification has been performed to induce apical closure when the pulp of immature permanent teeth is infected [6], but it cannot maintain pulp vitality [7]. Since the 1960s, regenerative endodontic procedures (REPs) have been proposed to be used in uninfected, especially traumatic pulp tissue to replace the infected/inflamed pulp tissue with viable tissue. Nevertheless, REPs have also been successfully used to treat necrotic pulps and immature apices, with or without apical periodontitis [8,9].

The dental pulp is a neural-crest-derived, highly specialized mesenchymal tissue that comprises odontoblasts and cells that produce extracellular matrix (ECM). The two updated strategies for REPs to regenerate dental pulp-like tissues are cell transplantation (cell-based) and cell homing (cell-free) [1]. The first strategy requires exogenously transplanted stem cells (SCs) to form the dentin/pulp-like complex in the subcutaneous connective tissue after transplant of stem cells, including postnatal human dental pulp stem cells (hDPSCs) [10], stem cells from human exfoliated deciduous teeth, periodontal ligament stem cells, dental follicle progenitor stem cells, and stem cells from apical papilla (SCAPs) [11]. The second strategy uses the host’s endogenous cells originated from the apical papilla to regenerate tissue, which may be more clinically translatable [1]. hDPSCs have the capacity to differentiate into multiple cell types, including odontoblasts, osteoblasts, and chondrocytes, by expressing specific markers, promoting alkaline phosphatase (ALP) enzyme activity, and producing precipitated mineralized nodules [12]. Basic research is aiming to establish more effective regenerative methods for hPDSC transplantation into root canals. Therefore, to circumvent root canal decontamination problems, maintain the vitality of pulp, and regenerate pulp-like tissue, researchers are seeking strategies to regenerate pulp-like tissue through either cell therapy or tissue-engineering methods.

Laser-induced photobiomodulation therapy (PBMT) has been proposed as an adjunctive therapy with the potential to improve dental pulp tissue regeneration [13]. Notably, Marques et al. (2016) [14] identified PBMT as “the fourth element of tissue engineering along with stem cells, scaffolds, and growth factors” because of its benefits properties, which are able to overcome some drawbacks of tissue engineering. When applied with adequate parameters, PBMT stimulates cell proliferation and differentiation [15], ATP production [16], mitochondrial respiration [17], protein synthesis, and bone formation in human periodontal ligament stem cells, fibroblasts, and odontoblasts [18], which are directly involved in tissue repair [18,19]. It has also been demonstrated that DPSCs respond positively to laser phototherapy, indicating that PBMT may be a crucial therapy for tissue engineering associated with stem cells [20]. As a matter of fact, the possible cell sources for pulp regeneration through cell homing include DPSCs, SCAPs, and bone marrow stem cells (BMSCs) [21].

The term “PBMT” is used to characterize the various laser/LED light applications with low-energy densities and is based on photochemical mechanisms where the photo energy is absorbed by the mitochondrial chromophores and transmitted to respiratory-chain components [22]. PBMT is activated through an electromagnetic radiation source and has been demonstrated in many clinical applications to exert anti-inflammatory, analgesic, and trophic-regenerative effects [22,23]. Previous studies have illustrated that in endodontics, PBMT has been widely accepted for its beneficial effects, such as analgesia, sterilization/disinfection, reduction of dentin sensitivity, and transpiration of infected dentin and dentin formation in root canals [24]. The red and near-infrared light emitted by PBMT is absorbed by the mitochondrial respiratory chain, which is one of the main sources of reactive oxygen species (ROS), thereby causing the production of ROS, nitric oxide, adenosine triphosphate (ATP), and cyclic adenosine monophosphate (cAMP) to initiate stem cell proliferation and induce the signal cascade effect [25]. Lasers, such as diode lasers, are usually used for PBMT of DPSCs, and the light is absorbed by mitochondria, causing metabolic changes in the host cells through a cascade of photochemical and photoelectric reactions, inducing both primary and secondary effects on the irradiated tissue [23].

Despite several basic and clinical studies being conducted on the effects of PBMT in endodontic treatments, a lack of understanding regarding PBMT’s molecular and cellular mechanisms in REPs, the interactions that occur after PBMT in the PDSCs, and the contradictory results of previous studies, have made it challenging to obtain a precise mechanism profile of PBMT. Accordingly, the purpose of the present review is to provide a knowledge profile by understanding the mechanisms used for cellular modulation of novel adjuvant treatment strategies in endodontics, specifically REPs, based on laboratory and clinical studies published until now.

## 2. The Application of PBMT in Endodontics

PBMT applied with a low-level laser (LLLT) provides endodontists with a non-invasive and non-thermal method that can be utilized as an adjunct to traditional RCT or as a therapeutic tool in REPs due to its anti-inflammatory effects, apical cicatrization [26], and acceleration benefits [27,28,29,30]. Commonly used clinical applications, such as direct pulp capping (DPC), treatment of dentine hypersensitivity, dental analgesia, and reduction of postoperative pain after endodontic treatment, have been investigated (Table 1). It is also an effective diagnostic tool for caries and pulp hyperemia.

### 2.1. PBMT-Induced Anesthesia

Being a non-invasive method, PBMT is able to produce anesthesia with an estimated significance of 60–95% [31]. In this regard, PBMT using an 810 nm diode laser (250 mW, 53.3 J/cm^2^ per side, 120 s, and continuous mode) achieves good-quality anesthesia during conventional tooth excavation [32]. In another study, PBMT based on the 2940 nm Er:YAG laser (60 mJ/point in non-contact mode) was used to achieve an appropriate anesthetic effect during Er:YAG-assisted cavity preparation of primary teeth [31]. Previous systematic reviews have shown that the current clinical parameters in this field are inadequate, so it does not seem feasible to propose a precise treatment protocol.

### 2.2. Laser-Assisted Diagnostics of Initial Caries Lesions and Pulp Status

Approaches based on light-induced fluorescence or light scattering properties related to demineralization are laser fluorescence (LF) [33], quantitative light-induced fluorescence (QLF) [34], and optical coherence tomography (OCT) [34], each of which is regarded as a useful supplement to the conventional diagnostic tools for caries and has helped improve the accuracy of caries detection in recent decades.

The LF technique, which is applied using a diode laser at a wavelength of 655 nm, quantifies the fluorescence intensity of a tooth’s surface and displays it. QLF detects the auto-fluorescence by irradiating teeth at 405 nm. OCT uses a 1300 nm wavelength to detect the light backscattered from tooth structures for dental caries detection applications. LF and QLF methods can quantify the severity of demineralization by measuring the fluorescence loss; however, the disadvantage of these methods is that they cannot measure the internal extension of carious lesions.

The patient will experience severe pain when the laser is used on a tooth with hyperemic pulp because laser irradiation increases the local blood flow in the pulp [35].

### 2.3. Laser-Based Prevention and Preparation of Enamel Caries

Diode lasers containing 810, 830, and 890 nm were used for caries prevention [36]. PBMT (810 nm, 30 mM, and 90 s) has been shown to increase calcium and phosphate levels [37,38]. PBMT applied at a wavelength of 830 nm suppressed the process of demineralization around orthodontic brackets on bovine teeth [39] and increased the hardness of the enamel surface [40].

The erbium laser is now also one of the options for cavity preparation, and it causes minimal invasive damage [41].

### 2.4. PBMT-Assisted Direct Pulp Capping

PBMT has been proposed to contribute to the outcomes of DCP procedures [42] and effectively improve the prognosis of DPC for permanent teeth [43], due to its considerable effects in shortening the inflammatory phase, reducing pain, promoting the process of wound healing, and stimulating the formation of hard dentin tissue [44,45]. The photo energy that penetrates into the pulp tissue coagulates the exposed pulp, thereby creating the biological basis for forming reparative dentin [12]. However, the findings of these in vitro and animal studies have shown that proper technique and materials (calcium hydroxide –Ca(OH)_2_ and MTA [46]) in DPC are prioritized over optimization of PBMT [47], and so far, laboratory results have not been generalized to clinical studies [44,48,49]. Therefore, it is still unclear how to identify the contribution of a laser’s application to the clinical-outcome improvement in the irradiated group, and further studies are needed to obtain more accurate results.

### 2.5. Decontamination of a Root Canal System

Due to infections by multiple and numerous aerobic and anaerobic bacteria, the key procedure of REPs comprises effective root canal decontamination (disinfection/sterilization). Decontamination of the root canal system is critical to the success of REPs and is accomplished through effective chemical-mechanical preparation, such as ultrasonically activated NaOCl [50]/EDTA [51] and the use of antibiotics in combination [52]. Aside from the ultrasonically activated NaOCl in endodontics, laser-assisted elimination of intra-canal microorganisms can be divided into two mechanisms: debris and smear-layer removal and disinfection of the root canal [53].

Maximum debris and smear-layer removal effects are achieved when laser light is used in root canals in conjunction with an appropriate concentration of a NaClO irrigating solution [53]. Many lasers, such as CO_2_ [54], Nd:YAG [55,56], and erbium family (Er:YAG [57] and Er,Cr:YSGG [58]) ones have been reported to be used in removing debris and smear layers from infected canal walls. Er:YAG is the most appropriate laser for this purpose [57,59].

Nd:YAG, Ho:YAG, and Er:YAG lasers eliminated more than 99% of *Enterococcus faecalis* (*E. faecalis*) and *Escherichia coli* (*E. coli*) for root-canal disinfection [60]. Schoop et al. (2004) demonstrated the antibacterial effects of diode, Er:YAG, Er,Cr:YSGG, and Nd:YAG lasers as being efficacious for dentinal disinfection from *E. facaelis* and *E. coli* at varying thicknesses. Therefore, laser treatment is a convenient adjunct to regular canal disinfection, especially in combination with chemical-mechanical preparations [61].

The anti-bacterial effects of photodynamic therapy (PDT) [62] on the pulp of human teeth with periapical and necrotic lesions indicated that it was an appropriate solution for root-canal disinfections [63]. Photodynamic therapy using a diode laser at 60 J/cm^2^ and 50 mW plus methylene blue eliminated *E. faecalis* from root canals to the degrees of 77% [64] and 99% [65]. Photodynamic therapy using diode red light at 30 J/cm^2^ energy plus methylene blue reduced 80% of the colonies of *Actinomyces israelii*, *Fusobacterium nucleatum*, *Porphyromonas gingivalis*, and *Prevotella intermedia*. Based on available studies [66], PDT provides endodontists with an antibacterial adjunctive device for RCT [67].

### 2.6. Postoperative Pain after Endodontic Treatment

Pain results from chemical-mechanical preparations or microbial damage to the pulp tissue or root apex, and its emergence is significantly higher after RCT [68,69,70]. This therapeutic technique, PBMT, makes endodontic treatment more comfortable by applying a preliminary phase to avoid the use of pharmacological agents for postoperative pain control [71].

PBMT using a 970 nm laser at 0.5 W reduced the postoperative pain after RCT in patients with symptomatic apical periodontitis [72]. However, another group showed that PBMT applied by an 808 nm laser with 100 mW of power at 70 J/cm^2^ has limited effects on reducing the pain associated with root-canal retreatment [68]. Thus, research has indicated that PBMT can delay the onset of postoperative pain and decrease its severity and duration after an RCT.

The mechanisms of PBMT’s actions in pain reduction are through facilitating the synthesis of anti-inflammatory prostaglandins (PGEs), immunoglobulins, β-endorphins, and lymphokines. Additionally, it decreases the production of pro-inflammatory factors and pain-related neurotransmitters. As a result, it has been suggested that PBMT could be useful in relieving pain after RCT or root-canal retreatment. However, due to a limited number of clinical studies, it is not yet time to formulate an exact clinical protocol; thus, further studies should be performed to achieve more conclusive results [73].

### 2.7. PBMT Used in Endodontic Surgery

Investigations have addressed the importance of PBMT in endodontic surgery with regard to pain relief, swelling reduction, and soft and hard tissue healing. The diode laser applied at 3–7.5 J/cm^2^ showed desirable results of PBMT on pain relief and tissue healing; however, more clinical studies are needed to obtain further insights. [26,74].

### 2.8. Tooth/Dentinal Hypersensitivity (DH)

Matsumoto et al. (2018) used a laser to treat DH for the first time, and laser technology is gradually being recognized as an important method for DH [75].

Up till now, lasers studied for DH treatment address three different mechanisms: dentin tubal obliteration by high-power density output laser therapy, alteration in the pain threshold of the pulp’s neural system, and stimulation of reactive dentine formation as a result of the PBMT effect [76,77]. Clinical studies have shown that PBMT using GaAlAs (795 or 830 nm) or InGaAlP (660 nm) at 1.8–10 J/cm^2^ significantly reduced DH.

### 2.9. Tooth Bleaching

PBMT mainly reduces the mild to severe postoperative sensitivity that appears in most patients after tooth bleaching [78], particularly with the in-office technique [79,80,81]. Clinical studies suggested that PBMT applied with a diode laser at 12 J/cm^2^ effectively reduced dental sensitivity after in-clinic bleaching. In vitro studies have investigated the effect of PBMT on odontoblastic cell responses or the neutralization of gel bleaching.

## 3. PBMT on Regenerative Endodontic Procedures

The application of lasers in REPs introduces the idea that PBMT induces biostimulatory effects on stem cells, including promoting stem cell growth, increasing their metabolism, improving their regeneration, accelerating dentine regeneration after pulp exposure [82,83], and having effective influences on the viability and differentiation of dentoalveolar-derived mesenchymal stem cells’ viability [84] (Figure 1). Pulp regeneration based on REPs under PBMT has shown favorable outcomes in several preclinical studies [13,85] and could be a feasible alternative to cell-homing therapies.

### 3.1. Biological Responses of hDPSCs to PBMT

The provoked bleeding is stimulated by mechanical forces on periapical tissues, which offer the essential regenerative elements of stem cells and scaffold to fill the canal space. These elements release growth factors to allow platelets to form a blood clot and stimulate cellular expansion [12]. Given that SCAPs with regenerative potential are found near the root apices in immature necrotic permanent teeth, this, combined with wide-open apices, facilitates the recruitment of more stem cells to the canal spaces, increasing the success of the REP modality [86]. Due to the survival and continued potential differentiation of the SCAPs, REO should be considered as the first choice of treatment for immature teeth with necrotic pulp [87].

The impact of PBMT on DPSCs has often been evaluated in terms of cell growth, survival rate, and cellular metabolism [88]. Moreover, the PBMT applied with 5 J/cm^2^ energy density presented the most striking results for maintaining cell viability, improving the proliferation and differentiation processes [21].

Additionally, upon stabilization of cell homing, the cells secrete the ECM containing growth factors, which are the third essential element of tissue engineering and thus are important for dental pulp regeneration. Garrido et al. (2019) showed that ECM secreted by hDPSCs exhibited a higher level of fibronectin when irradiated to the PBMT [89]. Thus, PBMT seems to help maintain and contribute to the balance of the cell homeostasis stabilization status.

Lovelace et al. (2011) studied in a rat model with pulp necrosis and an open apex the effects of PBMT on root development by using cell homing and stem-cell transplantation [90,91]. They found that daily PBMT irradiations improved the tissue’s response to apexification and favored apexogenesis, and thus played critical roles in maintaining the bio-stimulating effect during an REP [21].

Vascular endothelial growth factor regulates intercellular signals, the angiogenesis process, the formation of new blood vessels, and tissue regeneration [92]. This finding is supported by the results of Moreira et al. (2017), who reported that a blood-clot scaffold combined with PBMT resulted in the formation of dental pulp-like tissue with blood vessels, nerves, odontoblast-like cell layers, and perivascular SCs [13]. The continuous healing process of the pulp was observed in the presence of a blood clot as a scaffold, and the healing was accelerated in the PBMT-irradiated group [13].

Recently, SCs exposed to PBMT have demonstrated enhanced cell growth, which results in the activation of intracellular and extracellular chromophores and the initiation of cellular signaling [25]. Zaccara et al. (2020) showed that PBMT can regulate histone-acetylation signaling of hDPSCs through increasing the nuclear modifications that chemically induce histone acetylation on H3 (Lys9), and it can influence gene expression to increase the hDPSCs’ viability [93]. These results are consistent with studies that demonstrated that PBMT irradiation of hDPSCs using a 600 nm InGaAlP diode laser or a 635 nm LED laser was able to promote cell growth and survival, ATP production, and mitochondrial metabolic activity [94,95,96]. Moreover, Ferreira et al. (2019) reported that PBMT at 5 J/cm^2^ can help hDPSCs maintain undifferentiated status and replicate for a short-term period [97].

Regarding the influence of PBMT on the mineralization of hDPSCs, Matsui et al. (2007) irradiated dental pulp cells in vitro with a diode laser and observed significantly elevated expression of calcified nodules, higher ALP activity, upregulation of bone morphogenetic protein (BMP), and upregulation of osteocalcin after 1.0 W irradiation [98,99,100]. Additionally, PBMT can improve the composition of the extracellular matrix synthesized by the cell sheets of hDPSCs, facilitating cell transplantation by increasing fibronectin synthesis induced by PBMT [89].

Divergent results have been obtained based on in vivo and in vitro experiments about PBMT on regenerative pulp treatment procedures. Pereira et al. (2012) and Theocharidou et al. (2017) demonstrated that none of the PBMT protocols improved proliferation or cell viability rates, nor the relative production levels of mineralized nodules for hDPSCs from normal and inflamed dental pulps [101,102], whereas the latter showed that scaffold/DPSCs complexes irradiated by PBMT showed statistically significant increases in odontogenesis-related markers and ALP enzymic activity [102].

To clearly clarify the mechanisms of PBMT on hDPSCs, basic cytological and histological studies are needed. Nevertheless, current clinical studies strongly suggest that PBMT has positive effects on SCs migration, differentiation, proliferation, and cellular activity (Figure 2).

### 3.2. The Favorable Effect of PBMT on Vascularity and Fibroblast Proliferation

PBMT with 4 J/cm^2^ promotes significantly higher cell growth in terms of vascularity and fibroblast proliferation. Furthermore, PBMT influenced primary photochemical and photophysical activities in the mitochondria, resulting in rapid increases in ATP and cell viability due to an oxidative change. Additionally, a secondary PBM effect was triggered when biochemical reactions and redox state were changed, which led to DNA synthesis, and consequently, increased cell proliferation [82].

Moura-Netto et al. (2016) also reported increased proliferation of stem cells from exfoliated deciduous teeth (SHEDs) under these same PBMT parameters during situations of nutritional deficit [103]. This finding is consistent with those of Eduardo et al. (2008) and Marques et al. (2017) [94,104], who demonstrated that pulp fibroblasts from human primary teeth showed greater viability and proliferation when exposed to higher energy densities of lasers for a shorter period of time, introducing the hypothesis that in laser application, the dose applied plays a positive role in cell growth in vitro. One possible explanation is that the transitory heating of the chromophores that may occur over longer periods may trigger enzyme (cytochrome c oxidase) inhibition [105].

### 3.3. The Beneficial Effect of PBMT on Dentin Formation

PBMT stimulates dentin formation by inducing the dentin matrix to release a variety of growth factors, including BMP, fibroblast growth factors, and transforming growth factor-β, all of which stimulate dental pulp cell and odontoblastic differentiation, and are related to the ectodermal–mesenchymal molecular interactions [12].

For human primary teeth, the literature reports favorable tissue effects of PBMT for dentine formation [85,106] and immature-connective-tissue synthesis to fill the root canal [13]. PBMT increased the proliferation of human SHEDs [97] and maintained cell viability [97].

Fekrazad et al. (2015) [106] revealed that PBMT induces new dentine formation, and Arany et al. (2014) [85] showed that PBMT promoted hDPSCs and mouse pre-odontoblasts to differentiate new dentine and increased tertiary dentine volumes. Furthermore, Moreira et al. (2017) [13] observed fewer and thinner collagen fibers and blood vessels, and a layer of cells in intimate contact with the dentin wall that exhibited cytoplasmic extensions into the dentinal tubules in the PBMT-irradiated group, resulting in the formation of an immature connective tissue filling the mesial root canal and forming an odontoblast-like cell layer [13].

### 3.4. Current Limitations of PBMT in REPs

Although the studies mentioned before demonstrated the efficacy and success of PBMT for REPs, there are still many controversies about its effectiveness. Despite the variety of PBMT parameters and the different comparative methods used in clinical trials, it appears that accurate PBMT protocols need to be evaluated in well-designed and large clinical studies before achieving evidence-based treatment protocols and conclusions. Additionally, the combination of antimicrobial peptides (AMPs) with PBMT may be beneficial for the clinical REPs due to its potential immunomodulation properties. However, such combination therapy needs to be further investigated.

Although PBMT affects the SCs and pulp fibroblasts’ differentiation, proliferation, and viability—and dentine formation—the findings of existing in vivo and in vitro studies cannot yet fully explain the complete mechanisms of PBMT in REPs, and therefore, further prospective and randomized clinical trials on REPs’ efficacy with PBMT are required. In order to apply the therapy to patients, more in vivo research with adequately large samples must be carried out, even though the results until now have been promising for dentin–pulp complex regeneration.

Based on the limited evidence available to date, we hypothesize that PBMT has positive effects on regenerative endodontic procedures in terms of improved clinical outcomes or molecular modulations.

## 4. Conclusions

The present review indicates that PBMT-assisted regenerative pulp procedures could be useful adjunctive tools for future advancements in endodontics, specifically REPs. However, the currently available scientific evidence is unable to specifically explain the mechanism of action of PBMT in REPs. Therefore, further in vivo and in vitro research is required.

## Figures and Tables

**Figure 1 bioengineering-10-00371-f001:**
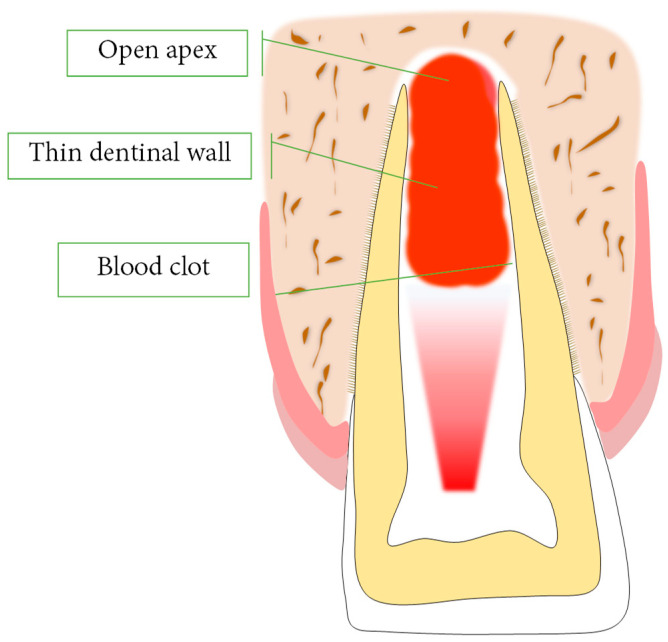
Scheme of a PBMT-assisted REP. The blood clot containing the stem cells, scaffolding, and growth factors is irradiated by PBMT to further induce favorable biostimulation.

**Figure 2 bioengineering-10-00371-f002:**
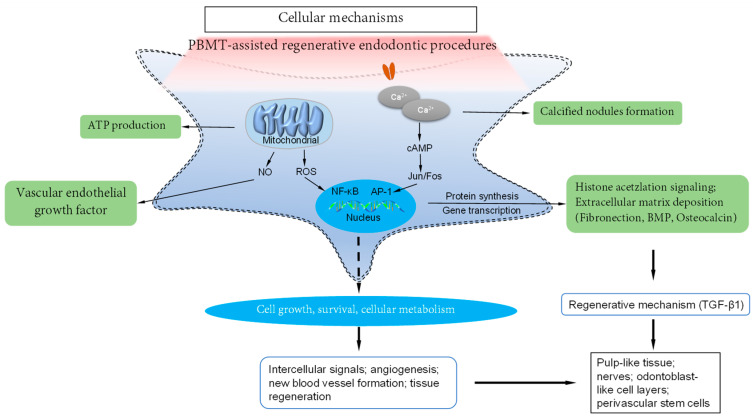
The cellular and molecular mechanisms of PBMT irradiation on DPSCs.

**Table 1 bioengineering-10-00371-t001:** Current applications of PBMT in endodontics.

**1**	PBMT-Induced Anesthesia
**2**	Laser-assisted Diagnostics of Initial Caries Lesions and Pulp Status
**3**	Laser-based Prevention and Preparation of Enamel Caries
**4**	PBMT-assisted Direct Pulp Capping
**5**	Decontamination of Root Canal System
**6**	Postoperative Pain after Endodontic Treatment
**7**	PBMT used in Endodontic Surgery
**8**	Tooth/Dentinal Hypersensitivity
**9**	Tooth Bleaching
**10**	Regenerative Endodontic Procedures

## Data Availability

The datasets used and/or analyzed during the current study are available from the corresponding author on reasonable request.

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
