# Peer review of "Photobiomodulation Therapy and Pulp-Regenerative Endodontics: A Narrative Review"

_bioengineering, 2023, doi:10.3390/bioengineering10030371_

Round 1

Reviewer 1 Report

The paper entitled “The Invaluable Application of Photobiomodulation Therapy on the Pulp Regenerative Endodontics: Molecular and Cellular Mechanisms” aims to provide a knowledge profile by understanding the mechanisms used for cellular modulation of the novel adjuvant treatment strategy in endodontics, specifically REP, based on laboratory and clinical studies published until now.

The article covers an interesting topic. I suggest some improvements before an eventual publication:

Major issues:

This is a narrative review and should be stated in the title. It does not add anything new to the scientific literature.

I suggest this title “Photobiomodulation Therapy and Pulp Regenerative Endodontics. A narrative review”

Minor issues:

please add the keywords. List three to ten pertinent keywords specific to the article yet reasonably common within the subject discipline.

Figure 1: the reviewer humbly thinks that Figure 1 is useless. A bullet list could be equally effective. Nevertheless this is left to the author’s opinions.

The authors could also cite similar findings by the following reviews:

Ansari G, Safi Aghdam H, Taheri P, Ghazizadeh Ahsaie M. Laser pulpotomy-an effective alternative to conventional techniques-a systematic review of literature and meta-analysis. Lasers Med Sci. 2018 Nov;33(8):1621-1629. doi: 10.1007/s10103-018-2588-4. Epub 2018 Jul 17. PMID: 30014215.

Vahdatinia F, Gholami L, Karkehabadi H, Fekrazad R. Photobiomodulation in Endodontic, Restorative, and Prosthetic Dentistry: A Review of the Literature. Photobiomodul Photomed Laser Surg. 2019 Dec;37(12):869-886. doi: 10.1089/photob.2019.4707. PMID: 31873065.

The authors could also support the advantages of the use of this kind of treatments in the pandemic era, when less appointments should be performed therefore less aerosol generating procedures could be selected. Please cite: Paolone G, Mazzitelli C, Formiga S, Kaitsas F, Breschi L, Mazzoni A, Tete G, Polizzi E, Gherlone E, Cantatore G. One-year impact of COVID-19 pandemic on Italian dental professionals: a cross-sectional survey. Minerva Dent Oral Sci. 2022 Aug;71(4):212-222. doi: 10.23736/S2724-6329.21.04632-5. Epub 2021 Dec 1. PMID: 34851068.

Author Response

Reviewer 1’s reports Major issues: Point 1. This is a narrative review and should be stated in the title. It does not add anything new to the scientific literature. I suggest this title “Photobiomodulation Therapy and Pulp Regenerative Endodontics. A narrative review” Response 1: Thank you very much for the great suggestion. We agree with the reviewer, and the title has been revised to “Photobiomodulation Therapy and Pulp Regenerative Endodontics: A Narrative Review”. (Please find the line 1-2 in the clean version of revised manuscript) Minor issues: Point 2. please add the keywords. List three to ten pertinent keywords specific to the article yet reasonably common within the subject discipline. Response 2: Thank you. We have added a keyword section with five pertinent keywords related to the present review’s topic. (Please find the line 39-41 in the clean version of revised manuscript) Point 3. Figure 1: the reviewer humbly thinks that Figure 1 is useless. A bullet list could be equally effective. Nevertheless this is left to the author’s opinions. Response 3: We appreciate the point raised by the reviewer. We agreed that the bullet list would be clear for the current application of PBMT in endodontics. Please find the Table 1. (Please find the line 126 in the clean version of revised manuscript) Point 4: The authors could also cite similar findings by the following reviews: Ansari G, Safi Aghdam H, Taheri P, Ghazizadeh Ahsaie M. Laser pulpotomy-an effective alternative to conventional techniques-a systematic review of literature and meta-analysis. Lasers Med Sci. 2018 Nov;33(8):1621-1629. doi: 10.1007/s10103-018-2588-4. Epub 2018 Jul 17. PMID: 30014215. Vahdatinia F, Gholami L, Karkehabadi H, Fekrazad R. Photobiomodulation in Endodontic, Restorative, and Prosthetic Dentistry: A Review of the Literature. Photobiomodul Photomed Laser Surg. 2019 Dec;37(12):869-886. doi: 10.1089/photob.2019.4707. PMID: 31873065. The authors could also support the advantages of the use of this kind of treatments in the pandemic era, when less appointments should be performed therefore less aerosol generating procedures could be selected. Please cite: Paolone G, Mazzitelli C, Formiga S, Kaitsas F, Breschi L, Mazzoni A, Tete G, Polizzi E, Gherlone E, Cantatore G. One-year impact of COVID-19 pandemic on Italian dental professionals: a cross-sectional survey. Minerva Dent Oral Sci. 2022 Aug;71(4):212-222. doi: 10.23736/S2724-6329.21.04632-5. Epub 2021 Dec 1. PMID: 34851068. Response 4: Thank you. These three pieces of literature are helpful for the present review, and we add the related content following the reviewer’s advice. (Please find the line 122 or reference list #28-30 in the clean version of revised manuscript) (Again, we would like to thank the reviewer 1 for the improvement in the manuscript quality.)

Reviewer 2 Report

This research is under the scope of this journal; the topic is relevant for readers, and this research deals with potentially significant knowledge to the field.

Introduction:

Page 2. Line 46 please add this reference: PMID: 21477154

Language needs improvement, especially in the Introduction and the Discussion. One of the major concerns here is the English. The authors should have the manuscript looked at for language and sentence composition. There are a lot of sentences in the manuscript that do not make sense because of the English. 

Discussion: please consider these references for RTC desinfection: PMID: 36681260 PMID: 26783392

Figure legends: Bad descriptions

There are many mistakes in the references section and in the text

Author Response

Reviewer 2’s reports

Point 1. Page 2. Line 46 please add this reference: PMID: 21477154

Response 1: Thank you. We add the content and reference in the context. (Please find the reference list #11 in the clean version of revised manuscript)

Point 2. Language needs improvement, especially in the Introduction and the Discussion. One of the major concerns here is the English. The authors should have the manuscript looked at for language and sentence composition. There are a lot of sentences in the manuscript that do not make sense because of the English.

Response 2: Thank you. According to the reviewer’s suggestion, we have improved the English language with the native speaker.

Point 3. Discussion: please consider these references for RTC desinfection: PMID: 36681260 PMID: 26783392

Response 3: Thank you. We have added these two references to the content. (Please find the reference list #62 and 66 in the clean version of revised manuscript)

Point 4. Figure legends: Bad descriptions. There are many mistakes in the references section and in the text

Response 4: Thank you, and now we have corrected the figure legend and reference list.

(Again, we would like to thank the reviewer 2 for the improvement in the manuscript quality.)

Reviewer 3 Report

The aim of this review was to present an overview of the application of laser as well as PBMT irradiation to the procedures of REP as well as in endodontics. This research is under the scope of this journal; the topic is relevant for readers, and this research deals with potentially significant knowledge to the field. And It will be important of Endodontics knowledge. The topic is relevant for readers and this study deals with potentially significant knowledge to the field and open new way for future studies. 

However, there are Major aspects which is need to be improved in the manuscript:

(Abstract)

  • please improve the abstract, based in the instructions for authors. The authors need to reformulate completely the abstract, and add more information the description of the search methodology used. Databases consulted (primary and secondary), search terms (keywords and Mesh terms), limits used, inclusion criteria, number of articles obtained and it evaluation.

(Keywords)

  • Please add keywords, and order the keywords / Mesh terms alphabetically

(Introduction) 

    • Which results are comparable with other studies? What has this study been new for dentistry?
    • The dental papilla, when the root is formed, is in the apical zone, called the papilla apical (stem cells from apical papilla - SCAPs) that remain until the apical closure of the root. Also these, cells can the survival of SCAPs at infection of the tooth, (in REPs animal study and a Clinical Case, read these references (doi.org/10.3390/APP9193942). HERS and the apical papilla are two embryologic structures which coordinate all the radicular development through epithelial–mesenchymal interactions. Apical papilla was a reservoir for mesenchymal stem cells (SCAPs) fundamental for root development of immature teeth. A normal dentin–pulp complex development requires not only the survival of HERS and ERM but also SCAP. So it seen very important to described also about REPs.
    • Add other PBMT in Vital Pulp therapy (in cases irreversible pulpite). Preoperative state of the pulp (inflamed versus non-inflamed): In the clinic, for this situation deep carious lesions, also we need more information to evaluate the impact of preoperative pulp inflammation. Please read this article https://doi.org/10.1016/j.joen.2021.06.018. In  this animal study, radiographic and histologic outcomes of full pulpotomy are not jeopardized by short-term preoperative pulp inflammation. The PBMT have also indication for irreversible pulpitis with full pulpotomy. What do you think about the difference between pulp inflammation versus pulp infection? And the individual immunological capacity to respond to the aggressor stimulus. Are these importance?
    • Other emerging filed are the Antimicrobial peptides (AMPs). AMPs was potencialy immunomodulation applications, such as LL37 peptides, may be immobilized nanoparticules (AuNPs) on the surface of scaffolds, to render them with antimicrobial and angiogenic properties and also interaction with behaviour of the Macrophages. Perhaps, authors may want to add AMP information applied with LASER.

Author Response

Reviewer 3’s reports

Point 1. (Abstract) Please improve the abstract, based in the instructions for authors. The authors need to reformulate completely the abstract, and add more information the description of the search methodology used. Databases consulted (primary and secondary), search terms (keywords and Mesh terms), limits used, inclusion criteria, number of articles obtained and it evaluation.

(Keywords) Please add keywords, and order the keywords / Mesh terms alphabetically

Response 1: Thank you. Now we have rewritten the abstract part and added the keyword section.

Point 2. (Introduction) Which results are comparable with other studies? What has this study been new for dentistry? The dental papilla, when the root is formed, is in the apical zone, called the papilla apical (stem cells from apical papilla - SCAPs) that remain until the apical closure of the root. Also these, cells can the survival of SCAPs at infection of the tooth, (in REPs animal study and a Clinical Case, read these references (doi.org/10.3390/APP9193942). HERS and the apical papilla are two embryologic structures which coordinate all the radicular development through epithelial–mesenchymal interactions. Apical papilla was a reservoir for mesenchymal stem cells (SCAPs) fundamental for root development of immature teeth. A normal dentin–pulp complex development requires not only the survival of HERS and ERM but also SCAP. So it seen very important to described also about REPs.

Response 2: Thank you. We agree that SCAPs are important for REPs, therefore, we describe the related content in this part. (Please find the line 276-278 in the clean version of revised manuscript)

Point 3. Add other PBMT in Vital Pulp therapy (in cases irreversible pulpite). Preoperative state of the pulp (inflamed versus non-inflamed): In the clinic, for this situation deep carious lesions, also we need more information to evaluate the impact of preoperative pulp inflammation. Please read this article https://doi.org/10.1016/j.joen.2021.06.018. In this animal study, radiographic and histologic outcomes of full pulpotomy are not jeopardized by short-term preoperative pulp inflammation. The PBMT have also indication for irreversible pulpitis with full pulpotomy. What do you think about the difference between pulp inflammation versus pulp infection? And the individual immunological capacity to respond to the aggressor stimulus. Are these importance?

Other emerging filed are the Antimicrobial peptides (AMPs). AMPs was potencialy immunomodulation applications, such as LL37 peptides, may be immobilized nanoparticules (AuNPs) on the surface of scaffolds, to render them with antimicrobial and angiogenic properties and also interaction with behaviour of the Macrophages. Perhaps, authors may want to add AMP information applied with LASER.

Response 3: Thank you for the reviewer’s good questions. We add the reference in the context. (Please find the reference list #46 in the clean version of the revised manuscript.)

We agree with the reviewer that AMPs combined with LASER may have the potential to be beneficial for the regerative pulp procedures. So, we add this to the discussion. (Please find the line 385-388 in the clean version of revised manuscript)

(Again, we would like to thank the reviewer 3 for the improvement in the manuscript quality.)

Reviewer 4 Report

The article is well written and scientifically sound, moreover gives a good overview of the field.

Some minor issue could be improved to give the readers a more comprehensive view 

The decontamination section could be expanded including the effects of different irrigation solution activated in different ways 

Di Spirito F, Pisano M, Caggiano M, Bhasin P, Giudice RL, Abdellatif D. Root Canal Cleaning after Different Irrigation Techniques: An Ex Vivo Analysis. Medicina 2022;58(2).

Lo Giudice G, Lizio A, Lo Giudice R, Centofanti A, Rizzo G, Runci M, et al. The effect of different cleaning protocols on post space: A SEM study. Int J Dent 2016;2016:1-7.

Considering the scarcity of articles in the field it could be interesting to expand the section of future prospective, and better descrive the main limitation found in the articles evaluated. 

The article should be defined in the text as narrative review.

Author Response

Reviewer 4’s reports

Point 1. The decontamination section could be expanded including the effects of different irrigation solution activated in different ways

Di Spirito F, Pisano M, Caggiano M, Bhasin P, Giudice RL, Abdellatif D. Root Canal Cleaning after Different Irrigation Techniques: An Ex Vivo Analysis. Medicina 2022;58(2).

Lo Giudice G, Lizio A, Lo Giudice R, Centofanti A, Rizzo G, Runci M, et al. The effect of different cleaning protocols on post space: A SEM study. Int J Dent 2016;2016:1-7.

Response 1: Thank you very much for the reviewer’s suggestion. We have added these two references in the context. (Please find the reference list #50 and 51 in the clean version of the revised manuscript.)

Point 2. Considering the scarcity of articles in the field it could be interesting to expand the section of future prospective, and better describe the main limitation found in the articles evaluated. The article should be defined in the text as narrative review.

Response 2: Thank you. We agree with the reviewer’s suggestion and the title has been revised with “Photobiomodulation Therapy and Pulp Regenerative Endodontics: A Narrative Review”.

(Again, we would like to thank the reviewer 4 for the improvement in the manuscript quality.)

Round 2

Reviewer 1 Report

All the corrections have been performed.

Author Response

Again, we would like to thank the reviewer 1 for the improvement in the manuscript quality.

Reviewer 3 Report

The authors improve the article after reviewers comments 

Author Response

Again, we would like to thank the reviewer 3 for the improvement in the manuscript quality.